# Starvation in Mice Induces Liver Damage Associated with Autophagy

**DOI:** 10.3390/nu16081191

**Published:** 2024-04-17

**Authors:** Katharina Schuster, Anna Staffeld, Annelie Zimmermann, Natalie Böge, Stephan Lang, Angela Kuhla, Linda Frintrop

**Affiliations:** 1Institute of Anatomy, Rostock University Medical Center, 18057 Rostock, Germany; katharina.schuster@med.uni-rostock.de (K.S.); annelie.zimmermann@med.uni-rostock.de (A.Z.); stephan.lang@med.uni-rostock.de (S.L.); 2Rudolf-Zenker-Institute for Experimental Surgery, Medical University Rostock, 18057 Rostock, Germany

**Keywords:** anorexia nervosa, murine model, murine liver, starvation-induced hyperactivity, running wheel activity, liver changes, hepatocyte injury, autophagy

## Abstract

Anorexia nervosa (AN) induces organ dysfunction caused by malnutrition, including liver damage leading to a rise in transaminases due to hepatocyte damage. The underlying pathophysiology of starvation-induced liver damage is poorly understood. We investigate the effect of a 25% body weight reduction on murine livers in a mouse model and examine possible underlying mechanisms of starvation-induced liver damage. Female mice received a restricted amount of food with access to running wheels until a 25% weight reduction was achieved. This weight reduction was maintained for two weeks to mimic chronic starvation. Alanine aminotransferase (ALT) and aspartate aminotransferase (AST) levels were measured spectrophotometrically. Liver fat content was analyzed using an Oil Red O stain, and liver glycogen was determined using a Periodic acid–Schiff (PAS) stain. Immunohistochemical stains were used to investigate macrophages, proliferation, apoptosis, and autophagy. Starvation led to an elevation of AST and ALT values, a decreased amount of liver fat, and reduced glycogen deposits. The density of F4/80^+^ macrophage numbers as well as proliferating KI67^+^ cells were decreased by starvation, while apoptosis was not altered. This was paralleled by an increase in autophagy-related protein staining. Increased transaminase values suggest the presence of liver damage in the examined livers of starved mice. The observed starvation-induced liver damage may be attributed to increased autophagy. Whether other mechanisms play an additional role in starvation-induced liver damage remains to be investigated.

## 1. Introduction

The hallmarks of the eating disorder anorexia nervosa (AN) are severe emaciation and hyperactivity [1]. Severe malnutrition induces widespread organ dysfunction associated with liver damage, such as hepatocyte injury [2]. This, in turn, can cause rises in the liver enzyme transaminases, namely alanine aminotransferase (ALT) and aspartate aminotransferase (AST), which can increase by up to 76% in AN patients [3,4,5,6]. These markers are commonly used to detect hepatocyte injury, as damaged hepatocyte membranes release liver enzymes into the bloodstream at high levels [7]. Starvation-induced hepatocyte damage may worsen the progress of AN. Therefore, in this study, we investigated the pathology of the liver due to starvation and the underlying mechanism of this liver injury.

The likely cellular mechanisms of liver damage due to starvation include (i) autophagic cell death, (ii) enhanced oxidative stress, (iii) increased apoptosis, and/or (iv) endoplasmic reticulum (ER) stress. The first mechanism, i.e., increased autophagy in hepatocytes, has been proposed based on the finding that many autophagosomes are present in the livers of AN patients [2,8]. Generally, cells respond to starvation with the process of autophagy, which involves the degradation of cytoplasmic constituents by engulfing a portion of the cytoplasm (reviewed in [9]). This allows the breakdown of molecules and the recycling of damaged or unnecessary components. In addition, another form of autophagic cell death called autosis, which is characterized by specific morphological features such as membrane damage, has been observed in the livers of individuals with AN [10]. However, it is unclear whether autophagy is the primary mechanism of liver damage due to starvation [11]. Apoptosis appears to play a subordinate role, as evidenced by no alteration in terminal deoxynucleotidyl transferase-mediated dUTP nick-end labeling (TUNEL)-positive hepatocytes [2]. Furthermore, liver biopsies of AN patients have shown an increase in ER stress, which is known to be influenced by nutrient availability [2]. Whether these mentioned mechanisms contribute to the clinical complications of eating disorders is not known.

To unravel this pathophysiology, we used the murine starvation-induced hyperactivity (SIH) model, which mimics the somatic consequences of AN, such as body weight loss, hyperactivity, and amenorrhea, as well as endocrine changes [12,13,14,15,16]. To mimic chronic starvation, a fixed body weight loss was induced, increasing the comparability of results [14]. To our knowledge, this is the first study systematically evaluating the morphological and functional hepatocyte changes in this model. Here, we aimed to evaluate the hepatic pathology and the underlying mechanism of hepatocyte injury during starvation. Therefore, we analyzed whether acute and chronic starvation affected the liver fat content, transaminase levels, proliferation or apoptotic rates, and autophagy.

## 2. Materials and Methods

### 2.1. Animals

Female C57BL/6J mice (*n* = 59, 4/8 weeks old) were purchased from Janvier Labs (Le Genest-Saint-Isle, France) as part of a large cohort study [16]. The mice were housed individually under a 12/12 h light/dark cycle with a light phase beginning at 6 AM, an air exchange rate of 355 m^3^/h, and a temperature of 22 ± 2 °C. Since AN predominantly affects females [17], we conducted our experiments using only female mice. Microbiological monitoring was implemented in accordance with the recommendations of the Federation of European Laboratory Animal Science Associations (FELASA), and the cages were changed once a week. The experiments were approved by the Review Board for the Care of Animal Subjects of the district government of Mecklenburg–Western Pomerania (reference number: 7221.3-1-005/21, approval date: 12 April 2021).

### 2.2. Study Design

The structure of the SIH paradigm (previously also called the modified activity-based anorexia (ABA) model) was described by Frintrop et al. [14] and Staffeld et al. [16]. During the acclimatization phase, the mice were provided ad libitum water and food for ten days. On the first day of the experiment, the mice were randomly assigned to either the different SIH (SIH_acute, SIH_chronic) or control group (Control_acute, Control_chronic). Body weight and food intake were evaluated daily at 1 PM. At this time point, the mice were also fed, and their running distance was measured every hour using running wheel sensors with integrated revolution detection and evaluated with VitalView Activity 1.4 software from STARR Life Science Corp (Oakmont, PA, USA). Standard chow (Ssniff, Soest, Germany) was fed to both experimental groups.

The acute starvation phase was defined as the six-day-long start of the starvation phase, in which the mice received 40% of their average daily food intake during the acclimatization phase until they reached a 25% weight loss (SIH_acute: *n* = 20). The average daily food intake was calculated based on the average daily food intake during the acclimatization phase, and then 40% of this average amount was determined. During both the acute and chronic starvation phases, the SIH animals were given unrestricted access to the reduced amount of food. Cages were checked for potential remaining chow on the following day before the SIH animals were fed again, but the animals always finished the limited amount of chow. Afterward, daily adjustments to food intake allowed for maintaining the 25% weight loss. To induce chronic starvation, an additional two weeks of starvation were included after the one-week acute starvation phase (SIH_chronic: *n* = 19). To maintain the 25% weight loss, the mice received 45–70% of their average daily food intake. If the body weight deviated by more than 2.5%, the food was increased or reduced in increments of 5%. Control mice were housed under the same conditions, except that they had ad libitum food access for the entire experiment (Control_acute: *n* = 10; Control_chronic: *n* = 10).

### 2.3. Locomotor Activity Determination

Each mouse was housed individually in a cage with a running wheel from STARR Life Science. Running wheel access was unlimited throughout the whole experiment. Running wheel activity (RWA) was evaluated using VitalView Activity 1.4 software from STARR Life Science Corp. The running activity was quantified by the number of revolutions per hour measured with a digital magnetic counter, which was combined with a microprocessor.

### 2.4. Biochemistry

Blood was collected by a retro-orbital blood draw at the end of the experiment. The glucose meter Contour^®^XT (Bayer, Leverkusen, Germany) was used to measure blood glucose levels directly after blood collection according to the manufacturer’s instructions. After centrifugation of the blood samples at 12,000 revolutions per minute (rpm) for 15 min at 4 °C, the supernatant was extracted, and serum samples were frozen at −80 °C. Spectrophotometric evaluation of ALT and AST levels was conducted as markers of hepatocellular damage using the cobas^®^c111Analyzer (Roche Diagnostics GmbH, Penzberg, Germany). The extinction at 340/378 nm was evaluated.

### 2.5. Histology, Immunohistochemistry, and Image Analysis

For liver preparation, the mice were injected intraperitoneally with ketamine (100 mg/kg) and xylazine (10 mg/kg). Then, they were transcardially perfused with phosphate-buffered saline (PBS) and 3.7% paraformaldehyde solution (pH 7.4). The whole liver was weighed, and the right superior and right inferior liver lobes were rapidly removed. The right superior lobes were processed and embedded in paraffin (Carl Roth, Karlsruhe, Germany), cut into 5 µm thick sections, and stained with hematoxylin–eosin staining (HE), PAS stain, or used for immunohistochemistry. The right inferior liver lobes were post-fixed with a paraformaldehyde solution overnight, incubated in PBS, and subsequently cryoprotected by immersion in 10%, 20%, and 30% sucrose in PBS at 4 °C. The liver lobes were embedded in an optimal cutting temperature medium (Sakura, Tokyo, Japan) and stored at −20 °C until further processing. Next, we made a series of 10 μm sections using a cryostat for analyzing liver fat content using Oil Red O staining, a dye used for staining neutral lipids such as triglycerides.

The histochemical procedures were performed using standard techniques as previously described in detail [18,19,20]. Briefly, the sections were incubated with 5% goat serum (Sigma, Munich, Germany) for 1 h, followed by overnight incubation at 4 °C. A list of antibody characteristics is provided in Table 1. Negative controls were performed by omitting primary antibodies or using monoclonal antibodies’ isotype controls. Then, the sections were incubated with 0.35% hydrogen peroxide (H_2_O_2_) in PBS for 30 min. The sections were incubated with the appropriate secondary antibodies, followed by the peroxidase-coupled avidin–biotin complex (ABC complex, Vector Laboratories, Newark, CA, USA). These two steps were replaced by the EnVision system (goat anti-rabbit, RRID: AB_2630375, catalog number: K4003, Dako, Hamburg, Germany) for the LC3B staining. The antigenic sites were detected by a reaction with 3,3’-diaminobenzidine (Dako).

TUNEL staining was performed using the ApopTag in situ apoptosis detection kit (RRID: AB_2661855, catalog number: S7100, Chemicon International, Temecula, CA, USA) according to the manufacturer’s protocol with minor modifications. After deparaffinization, sections were incubated with proteinase K (catalog number: EO0491, Thermo Fisher Scientific) at 20 µg/mL for 15 min. This was followed by incubation in 3% H_2_O_2_ in PBS for 5 min and incubation in an equilibration buffer for 15 min. Sections were then treated with the diluted terminal deoxynucleotidyl transferase (TdT) enzyme for 1 h at 37 °C. After washing with stop buffer, the sections were incubated in anti-digoxigenin conjugate for 30 min. Visualization of TUNEL-positive nuclei was then performed using 3,3′-diaminobenzidine (Dako).

Hematoxylin (Merck, Darmstadt, Germany) and eosin (Merck) staining and PAS staining (Diapath, Martinengo, Italy) were performed using standard protocols. The sections were digitized using the Grundium Ocus^®^ 40 digital slide scanner (Grundium, Tampere, Finland) and the Leica DM6 B automated microscope, equipped with a DMC6200 camera (Leica Microsystems CMS GmbH Wetzlar, Germany; 20-fold objective, numerical aperture (NA): 0.55; 40-fold objective, NA: 0.95).

We measured the microvesicular fat content using an Oil Red O stain on cryosections by generating a z-stack scan of every 20th region of the liver. Background subtraction was performed using ImageJ’s (version 1.48v, NIH, Bethesda, MD, USA) rolling ball algorithm [21], and then the area covered by red staining particles was calculated with the software ImageJ. The results were transformed into percentage area values and are shown as staining intensity as a percentage of the entire region of interest (ROI).

PAS staining was analyzed using thresholds in the software ImageJ. The results are presented as percentages of the stained area.

For the analysis of proliferating cells and macrophage numbers, the liver scans were overlaid with a grid with 250 µm grid spacing using the software QuPath (version 0.3.2) [22], and every 25th square was selected for cell counting. The cell counting was carried out by two evaluators who were blinded to the experimental groups. To determine the cell proliferation index, the number of KI67^+^ cells was divided by the total number of cells in the selected regions. The analysis of the LC3B^+^ staining was performed on every 25th square of the liver section using the software ImageJ. The number of TUNEL^+^ cells was counted by two blinded evaluators by examining the entire liver section.

### 2.6. Statistics

The data are illustrated as means and standard errors of the mean (SEM). The data for body weight and running activity were evaluated for acclimatization (days 1–10), acute starvation (days 11–16), and chronic starvation phase (days 17–29). The evaluation of the parameters of body weight and running activity between SIH and control mice within each phase of starvation was performed by a two-way ANOVA with repeated measurements. This ANOVA was used to compare the body weight and RWA between SIH and control mice for the duration of phases (phase of acclimatization, acute starvation phase, or chronic starvation phase). For post hoc comparisons, Bonferroni’s test was used to compare the control and SIH groups. The significance level was set to 5%. All data were tested for normality by performing the Shapiro–Wilks test. The Mann–Whitney test was then used to analyze liver fat contents, AST and ALT levels, numbers of KI67^+^ cells, and TUNEL^+^ cells. The parameters of liver weight, blood glucose, PAS staining, and LC3B^+^ staining were analyzed by Student’s *t*-tests. Macrophage numbers were tested for normality by the Shapiro–Wilks test and then analyzed using Welch’s *t*-test. All evaluations were performed using SPSS version 20 (IBM, Chicago, IL, USA).

### 2.7. Literature Research

A literature analysis was conducted using the search terms liver and anorexia nervosa, as well as transaminases and anorexia nervosa, in the PubMed database.

## 3. Results

### 3.1. Acute and Chronic Starvation Lead to Weight Loss and Hyperactivity

First, we analyzed whether the induced body weight loss in acute and chronic starvation leads to AN-related symptom hyperactivity (Figure 1). A total of 25% body weight loss was achieved after 5–6 days of starvation in the SIH mice during the acute starvation phase (SIH_acute: 5.9 days of starvation ±0.2, SIH_chonic: 4.7 days ± 0.3, Figure 1A, see red dotted line). The body weight of SIH mice was significantly decreased during acute and chronic starvation (acute starvation phase: Control_acute: 18.07 g ± 0.38 vs. SIH_acute: 15.59 g ± 0.26, *p* ≤ 0.001; Control_chronic: 18.07 g ± 0.5 vs. SIH_chronic: 14.9 g ± 0.4, *p* ≤ 0.001; chronic starvation phase: Control_chronic: 19.9 g ± 0.5 vs. SIH_chronic: 13.5 g ± 0.3, *p* ≤ 0.001, Figure 1A).

Additionally, running activity was significantly increased in SIH mice during acute and chronic starvation compared to the control group (acute starvation phase: Control_acute: 8.81 km ± 0.85 vs. SIH_acute: 10.81 km ± 0.63, *p* ≤ 0.05; Control_chronic: 7.13 km ± 0.67 vs. SIH_chronic: 9.5 km ± 0.64, *p* ≤ 0.001; chronic starvation phase: Control_chronic: 6.85 km ± 0.79 vs. SIH_chronic: 9.21 km ± 0.69, *p* ≤ 0.001, Figure 1B). In summary, acute and chronic starvation led to AN-related symptoms of hyperactivity.

### 3.2. Starvation Can Induce Morphological Liver Damage, Reduced Liver Glycogen Deposits, and Reduced Liver Fat Content

Assessment of liver histology using HE stains showed regions with hepatocyte damage and signs of inflammation in several SIH mice (Figure 2A). Glycogen deposits, hepatic mucins, and mucopolysaccharides stained by PAS staining were reduced in SIH mice compared to the control animals (acute starvation phase: Control_acute: 42.78% ± 0.76 vs. SIH_acute: 36.94% ± 2.10, *p* ≤ 0.05; chronic starvation phase: Control_chronic: 43.98% ± 0.88 vs. SIH_chronic: 37.03% ± 1.37, *p* ≤ 0.05, Figure 2B,C).

Acute starvation did not induce a significant change in the expression of Oil Red O (Control_acute: 3.45% ± 0.28 vs. SIH_acute: 3.10% ± 0.62, *p* = 0.06), while chronic starvation led to a decrease in Oil Red O regions in the livers of SIH mice (Control_chronic: 2.91% ± 0.37 vs. SIH_chronic: 1.80% ± 0.10, *p* ≤ 0.01, Figure 2D,E). In summary, starvation led to morphological liver damage in several animals, as well as a decrease in liver glycogen and mucins. Chronic starvation induced a decrease in liver fat content.

### 3.3. Starvation Affects the Liver, Resulting in Increased Serum Transaminases and Decreased Proliferation Rate

The liver weights of SIH mice were significantly reduced compared to the corresponding control (Control: 1.44 g ± 0.04 vs. SIH: 0.99 g ± 0.04, *p* ≤ 0.001, Figure 3A). In addition, the blood glucose levels in SIH mice decreased after starvation, indicating that the glycogen liver stores were depleted (Control: 152.4 mg/dL ± 5.44 vs. SIH: 102.54 mg/dL ± 4.99, *p* ≤ 0.001, Figure 3B). The investigation of serum transaminases revealed that ALT and AST levels in SIH mice increased significantly after starvation, indicating hepatocyte injury (ALT: Control: 45.37 U/L ± 1.63 vs. SIH: 73.43 U/L ± 7.98, *p* ≤ 0.05, AST: Control: 65.37 U/L ± 2.76 vs. SIH: 144.92 U/L ± 16.27, *p* ≤ 0.001, Figure 3C).

Furthermore, acute and chronic starvation led to a decrease in the KI67 proliferation index in SIH mice compared to the controls, indicating decreased liver proliferation (acute starvation: Control_acute: 3.39% ± 0.42 vs. SIH_acute: 0.92% ± 0.24, *p* ≤ 0.001, chronic starvation: Control_chronic: 4.54% ± 0.31 vs. SIH_chronic: 1.24% ± 0.13, *p* ≤ 0.001, Figure 3D). Analysis of apoptosis using TUNEL stains showed no change in terms of the number of apoptotic cells after starvation. In summary, starvation-induced increases in serum transaminases were associated with a decrease in cell proliferation, while apoptosis remained unchanged.

### 3.4. Starvation Leads to a Reduction in Hepatic Macrophage Density and an Increase in Autophagy-Related Protein Staining

Acute and chronic starvation led to a significant decrease in F4/80^+^ cells in the livers of SIH mice, representing a reduction in myeloid cells due to starvation (acute starvation: Control_acute: 9.02% ± 0.63 vs. SIH_acute: 3.79% ± 0.36, *p* ≤ 0.001; chronic starvation: Control_chronic: 8.69% ± 1.28 vs. SIH_chronic: 4.9% ± 0.46, *p* ≤ 0.05, Figure 4A). Chronic starvation induced an increase in autophagy-related protein staining measured in the livers, indicating an elevated autophagosome density in SIH mice after a longer period of starvation (chronic starvation: Control_chronic: 1.03% ± 0.17 vs. SIH: 2.47% ± 0.32, *p* ≤ 0.01, Figure 4B).

## 4. Discussion

The eating disorder, AN, is accompanied by severe emaciation and extensive locomotor activity. AN is a disease that affects multiple organs. We focused on the starvation-induced changes in the liver, as this organ plays an important role in nutritional storage in conditions of malnutrition. To analyze the pathophysiology of the liver, we used a murine SIH model that mimics the core features of AN, such as body weight loss, hyperactivity, and amenorrhea [12]. In this study, both acute and chronic starvation induced hyperactivity in all groups, which might be an excessive drive to forage in mice. Further, the hyperactivity of SIH mice is paradoxical when considering that the liver as energy storage might be injured after chronic starvation; thus, we further evaluate the hepatic pathology during starvation.

Chronic starvation induces changes in the liver, such as a reduced liver fat content and a reduction in hepatic glycogen and mucins stained by PAS staining. These findings are consistent with observations of depleted glycogen stores in the livers of AN patients [23]. As short-term starvation did not result in significant changes, we suggest that there was a gradual depletion of liver fat stores due to the prolonged period of starvation. Overall, we did not find evidence of steatohepatitis or related inflammatory processes. However, sections with liver damage and signs of inflammation were found in a few starved animals. This suggests that a 25% body weight reduction affected some animals’ livers more severely than others.

In some cases, patients with AN have revealed severe liver complications, including acute liver insufficiency and micro- and macrovesicular steatosis [2,24]. Overall, steatohepatitis appears not to be the main cause of liver damage in both animal and human studies. In addition, hepatic steatosis was found in patients with AN during the refeeding process (reviewed in [25]), caused by a high amount of hepatic fat and glucose deposition [26]. Therefore, hepatic complications can be caused by starvation due to nutrient deprivation or refeeding due to the sudden reintroduction of nutrients (reviewed in [27]). Appendix A provides a summary of the findings concerning liver damage in AN patients. In summary, it appears that ALT and AST vary in patients with AN, which might be due to differences in the degree of hepatic damage. Body Mass Index (BMI) might influence the hepatic findings in human AN. Thus, the role of the extent of starvation due to hepatic failure should be further analyzed in animal models.

The depletion of glycogen stores is suggested to be the main reason for the low blood glucose levels in starved animals, as was previously proposed in patients with AN [3]. In the current study, increases in ALT and AST levels were observed, indicating hepatocyte injury, which is consistent with findings in human studies [4]. The increased ALT and AST levels were normalized after weight rehabilitation [2]. Our study showed that the proliferation in the livers of starved animals was reduced, which may be due to energy depletion in cells. This is concurrent with results by Tessitore, which show a decrease in cell proliferation in rat livers after four days of starvation [28]. Additionally, chronic starvation led to a decrease in macrophage density, which contradicts the involvement of inflammation.

Potential mechanisms underlying liver damage due to starvation include (i) autophagic cell death, (ii) enhanced oxidative stress, (iii) increased apoptosis, and/or (iv) ER stress. To the best of our knowledge, this is the first time we have demonstrated that starvation causes an increase in autophagy-related protein staining in the livers of SIH mice. Previous findings have revealed increased autophagy in the hypothalamus of ABA animals, further highlighting the importance of this process in response to starvation [29]. The ABA model involves a period of ad libitum food intake for 1–3 h per day and free access to a running wheel. Another model investigating autophagy in GFP-LC3 transgenic mice could show an increase in autophagy in several tissues after 24 and 48 h of starvation [30]. Furthermore, electron microscopy studies have revealed the presence of many autophagosomes as well as a decrease in the density of organelles in liver biopsies from patients with AN [2,8]. It is known that starvation-induced autophagy is influenced by glucagon [31,32], amino acids [33,34], and insulin [33], i.e., glucagon stimulates autophagy and amino acids are inhibitors of autophagy (reviewed in [8]). Another potential mechanism is enhanced oxidative stress, as evidenced by food restriction in rats, which induced a reduction in total antioxidant status (TAS) [35]. In our study, we did not observe a change in apoptosis by the TUNEL assay, consistent with the findings of Rautou et al., who demonstrated no change in hepatic apoptosis after starvation [2]. However, their results showed hints of ER stress with an analysis of organelle morphology using electron microscopy [2]. Overall, increased apoptosis appears to play a subordinate role, while oxidative stress and ER stress might be contributing factors. As unfolded protein response (UPR) signaling following ER stress can induce autophagy [36,37], these two mechanisms might interact during periods of starvation. In summary, starvation-induced autophagy appears to be a principal mechanism involved in liver cell damage in animals and humans. To separate the effects of hyperactivity and starvation, in future studies, one control group without running wheels and one starvation group without running wheels will be included.

## 5. Conclusions

The depletion of liver fat deposits and the increase in autophagy are associated with chronic starvation, while the other analyzed aspects are similarly influenced by both acute and chronic starvation. Our study shows that liver damage in the SIH model of chronic starvation appears to be associated with increased autophagy. Whether autophagy is the primary underlying mechanism of starvation-induced liver injury and if the presented liver changes are reversible after refeeding requires further study.

## Figures and Tables

**Figure 1 nutrients-16-01191-f001:**
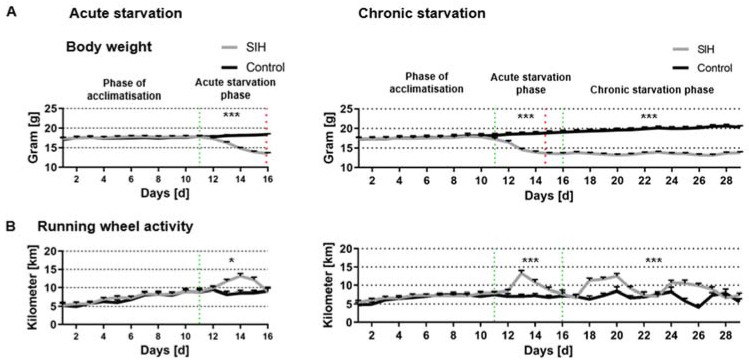
Acute and chronic starvation induce hyperactivity. (**A**) Body weight and (**B**) running wheel activity during acute starvation (**left**) and chronic starvation (**right**) in SIH (*n* = 39) and control mice (*n* = 20). The parameters of body weight and running activity were measured daily. The green lines separate the different starvation phases. The red lines represent that the SIH mice achieved a 25% weight reduction. * *p* ≤ 0.05, *** *p* ≤ 0.001, two-way ANOVA with repeated measurements.

**Figure 2 nutrients-16-01191-f002:**
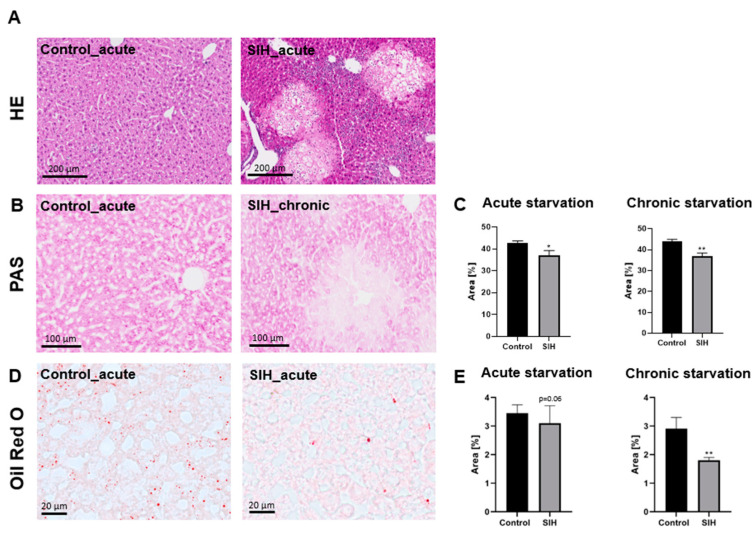
Representative section of region with visible liver damage compared to liver histology of a control animal (**A**). Representative sections of PAS staining (**B**) and quantification of stained area (**C**). Representative sections of Oil Red O stainings (**D**) and liver fat quantification in control (*n* = 19) and SIH mice (*n* = 39) (**E**). * *p* ≤ 0.05, ** *p* ≤ 0.01, (**C**) two-sided Student’s *t*-test, and (**E**) Mann–Whitney test.

**Figure 3 nutrients-16-01191-f003:**
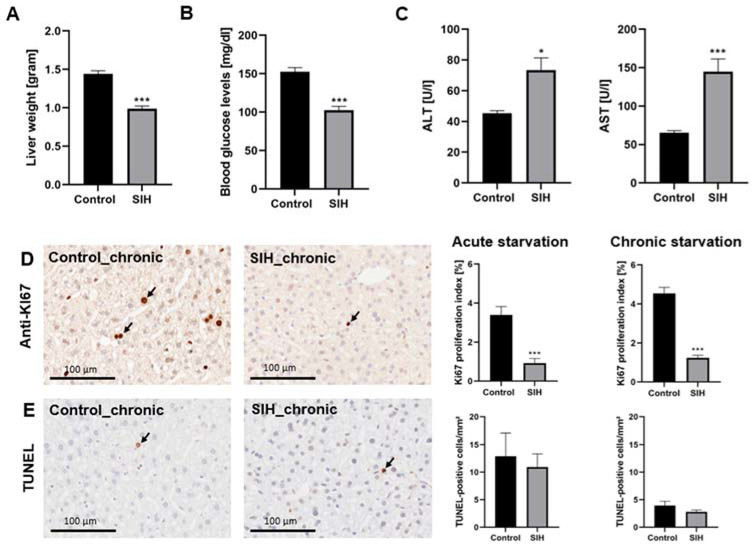
Starvation leads to decreased liver weight, decreased blood glucose, increased serum transaminases, and reduced KI67^+^ cells. (**A**) Liver weight, (**B**) blood glucose, and (**C**) serum ALT and AST levels in control (*n* = 19) and SIH mice (*n* = 37) after starvation. These findings are presented by combining results from both acute and chronic starvation. (**D**) Representative sections of KI67 staining and quantification of KI67^+^ cells in control (*n* = 20) and SIH mice (*n* = 36) after starvation. Scale bar = 100 µm. (**E**) Representative sections of TUNEL staining and quantification of TUNEL^+^ cells in control (*n* = 20) and SIH mice (*n* = 36) after starvation. Scale bar = 100 µm. Arrows demonstrate exemplary positive cells. * *p* ≤ 0.05, *** *p* ≤ 0.001, (**A**,**B**) two-sided Student’s *t*-test, (**C**–**E**) Mann–Whitney test.

**Figure 4 nutrients-16-01191-f004:**
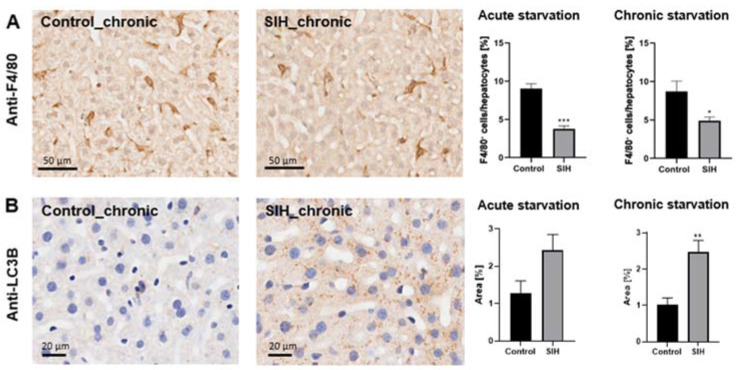
Acute and chronic starvations induce a reduced macrophage density and an increase in autophagy-related protein staining. Representative sections of (**A**) F4/80 and (**B**) LC3B staining and quantification in livers of control (*n* = 14) and SIH mice (*n* = 27) after starvation. * *p* ≤ 0.05, ** *p* ≤ 0.01, *** *p* ≤ 0.001, (**A**) Welch’s *t*-test and (**B**) two-sided Student’s *t*-test.

**Table 1 nutrients-16-01191-t001:** List of antibodies used for immunohistochemical stains.

Antigen	Species	Dilution	Purchase Number	RRID	Supplier
Primary Antibodies					
Ki67	Rabbit polyclonal	1:500	ab15580	AB_443209	Abcam, Cambridge, UK
F4/80	Rat monoclonal	1:100	MCA497	AB_2098196	Bio-Rad, Hercules, CA, USA
LC3B	Rabbit polyclonal	1:500	ab48394	AB_881433	Abcam, UK
Secondary Antibodies					
Anti-rabbit IgG	Goat	1:200	BA9400	AB_2313606	Vector Laboratories, USA
Anti-rat IgG	Goat	1:200	BA1000	AB_2336202	Vector Laboratories, USA
Isotype Controls					
IgG2b	Rat	1:100	02-9288	AB_2532966	Thermo Fisher Scientific, Waltham, MA, USA

## Data Availability

Data may be made available upon sending requests to the correspondence.

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
