# Peer review of "Starvation in Mice Induces Liver Damage Associated with Autophagy"

_nutrients, 2024, doi:10.3390/nu16081191_

Round 1
Reviewer 1 Report
Comments and Suggestions for Authors
1. some Keywords should be added, i.e. murine model or mice, autophagy
2. section 2.1. environmental air change ration should be added
3. section 2.2. more details on animal nutrition are needed, including what feed was used (company, composition) and in detail how feed intake was analysed, whether non-eating was taken into account
4. section 2.4. more details about blood sampling, survivor or post mortem
5. Results: the magnification scale should be indicated on all microscope images
6. references 44 and 45 were published more than 60 or 40 years ago, please change to more recent ones
Reviewer 2 Report
Comments and Suggestions for Authors
This study investigates the liver damage that occurs with anorexia nervosa in a mouse model and proposes that autophagy may be involved. The paper is well written, and the methods are clearly described, providing appropriate detail. I have a few minor comments.
The number of animals is provided in the methods section. Did all make it through? The number of animals should be shown in the figure legend.
Table 2 takes up a large amount of space in the paper, but only three sentences are dedicated to it. The table may be better as a supplement as it is not the paper's primary focus.
Figures 2, 3, and 4: Are the pictures from acute or chronically starved animals? Please add that to the figure legend.
Conclusion: The first line states that autophagy is associated with the length of starvation. This statement should be modified. Only two-time points were studied: acute and 14 days of chronic starvation. It was not a time-course study.
Supplementary Materials: Where is Table S1?
Institutional Review Board Statement: The study involved animals, not humans. IRBs review human studies. The words “Institutional Review Board Statement” should be changed to “Review Board for the Care of Animal Subjects:”.
A supplementary file with images was included. Where is this mentioned in the manuscript?
Reviewer 3 Report
Comments and Suggestions for Authors
In the paper entitled “Apoptosis and Cell Proliferation Are Involved in the Initiation of Liver Carcinogenesis by a Subnecrogenic Dose of Diethylnitrosamine in Refed Rats” Shuaster and co-workers focused their attention on the effects of anorexia nervosa induced starvation as first consequence of maltuntrition. This pathology is claimed to induce organ dysfunction that may lead to death. Authors conclude that starvation may induce hepatic autophagy that can be considered as a primary cause of liver damage. Despite the very interesting topic, some serious concerns are to be assessed to the Authors.
-The paper is well written, and figures are clear, nevertheless the rationale of this study results is not clear. --Abstract section needs to be revised, since rationale and results are not clear.
-Data included in these study results already published in different animal model. Authors should check out the paper entitled “In vivo analysis of autophagy in response to nutrient starvation using transgenic mice expressing a fluorescent autophagosome marker” by Noboru Mizushima and colleagues.
-Data included in the results section par. 3.3 has been demonstrated in a human model by Amanda Su and co-workers in the paper entitled “Two Acute Liver Injuries in a Patient with Malnutrition”.
-Authors conclude that liver damage is associated with autophagy linked with starvation inducing apoptosis, furthermore, these data have already published by Tessitore in other rodent animal model: “Apoptosis and cell proliferation are involved in the initiation of liver carcinogenesis by a subnecrogenic dose of diethylnitrosamine in refed rats”.
Round 2
Reviewer 1 Report
Comments and Suggestions for Authors
For the revised text of the manuscript, I have no further comments. Thank you
Reviewer 2 Report
Comments and Suggestions for Authors
Thanks for making the suggested changes. I have one more comment: The figures do not state whether the controls are from the acute or chronic groups.
Reviewer 3 Report
Comments and Suggestions for Authors
In the paper entitled “Starvation in Mice Induces Liver Damage Associated with Autophagy” Shuaster and co-workers focused their attention on the effects of anorexia nervosa induced starvation as first consequence of maltuntrition. This pathology is claimed to induce organ dysfunction that may lead to death. Authors conclude that starvation may induce hepatic autophagy that can be considered as a primary cause of liver damage. Authors in the revised version, improve the quality of the paper, answering to most the concerns of reviewers. Despite these efforts, some concern remains unclear. Authors presented very clear results in mice model regarding the role of starvation in liver damage, but they didn’t show new results, all data presented confirm, in a different animal model, results already published in literature. Authors are invited to show a mechanism involved in autophagy-starvation induced in liver to complete the observational results.
